# Imidacloprid disrupts larval molting regulation and nutrient energy metabolism, causing developmental delay in honey bee *Apis mellifera*

Zhi Li[1,2]*[†], Yuedi Wang[1,2][†], Qiqian Qin[1,2], Lanchun Chen[1,2], Xiaoqun Dang[1,2], Zhengang Ma[1,2], Zeyang Zhou[1,2,3,4]

[1]College of Life Sciences, Chongqing Normal University, Chongqing, China; [2]Key Laboratory of Pollinator Resources Conservation and Utilization of the Upper Yangtze River, Ministry of Agriculture and Rural Affairs, Chongqing, China; [3]Chongqing Key Laboratory of Microsporidia Infection and Control, Chongqing, China; [4]The State Key Laboratory of Resource Insects, Southwest University, Chongqing, China

**\*For correspondence:**
lizhicqnu@gmail.com

[†]These authors contributed equally to this work

**Competing interest:** The authors declare that no competing interests exist.

**Abstract** Imidacloprid is a global health threat that severely poisons the economically and ecologically important honeybee pollinator, *Apis mellifera*. However, its effects on developing bee larvae remain largely unexplored. Our pilot study showed that imidacloprid causes developmental delay in bee larvae, but the underlying toxicological mechanisms remain incompletely understood. In this study, we exposed bee larvae to imidacloprid at environmentally relevant concentrations of 0.7, 1.2, 3.1, and 377 ppb. There was a marked dose-dependent delay in larval development, characterized by reductions in body mass, width, and growth index. However, imidacloprid did not affect on larval survival and food consumption. The primary toxicological effects induced by elevated concentrations of imidacloprid (377 ppb) included inhibition of neural transmission gene expression, induction of oxidative stress, gut structural damage, and apoptosis, inhibition of developmental regulatory hormones and genes, suppression of gene expression levels involved in proteolysis, amino acid transport, protein synthesis, carbohydrate catabolism, oxidative phosphorylation, and glycolysis energy production. In addition, we found that the larvae may use antioxidant defenses and P450 detoxification mechanisms to mitigate the effects of imidacloprid. Ultimately, this study provides the first evidence that environmentally exposed imidacloprid can affect the growth and development of bee larvae by disrupting molting regulation and limiting the metabolism and utilization of dietary nutrients and energy. These findings have broader implications for studies assessing pesticide hazards in other juvenile animals.

## eLife assessment

This investigation of the changes in gene expression and some of the physiological consequences of sublethal exposures to the neonicotinoid insecticide imidacloprid in honeybee larvae is **useful**, although numerous experiments were not considered based on technical issues. The methodological design leads to concerns and it is therefore not obvious that all conclusions are justified. The study adds to our understanding of how this insecticide impacts development and growth of honeybees, but the evidence supporting the major claims is **incomplete**.

## Introduction

Hazardous pesticides are used extensively around the world to protect crop production, particularly in agriculture-dependent developing countries. However, exposure to these chemicals has become a global health concern. Imidacloprid, introduced in 1991, is known for its high animal toxicity and has received international attention (*Jeschke et al., 2011*). The World Health Organization and the U.S. Environmental Protection Agency have listed it as an acute oral and dermal toxicant for animals. The excessive use of imidacloprid can lead to environmental contamination, endangering human and animal health through food chains, poisoning non-target pollinators, and disrupting ecological balances (*Tudi et al., 2021*).

Imidacloprid is also known for its adverse effects on human health. Adults exposed to high doses of imidacloprid (9.6%) experience dizziness, apathy, movement disorders, labored breathing, and temporary growth retardation (*Wu et al., 2001*). Imidacloprid also causes adverse congenital, hematological, hepatic, and renal effects, along with degenerative changes in various organs (*Khan et al., 2010*; *Shaw et al., 2014*; *Yang et al., 2014*). Imidacloprid even at low doses causes severe liver, thyroid, and body weight problems, reproductive toxicity, developmental retardation, and neurobehavioral deficits in rats and rabbits (*Anatra-Cordone and Durkin, 2005*). Additionally, imidacloprid affects the singing ability of birds, making it difficult for them to attract mates and reproduce. It is a significant contributor to the decline in terrestrial insect populations since 1991, averaging 9% per decade (*Carneiro et al., 2022*; *Sánchez-Bayo and Wyckhuys, 2019*). Honey bees (*Apis mellifera* L.) are valuable model organisms in insects with significant economic and ecological importance as crop pollinators and in maintaining ecological balance (*Zheng and Fu-Liang, 2009*). However, the widespread use of pesticides, including imidacloprid, has resulted in colony collapse disorder, causing severe impacts on the beekeeping industry and ecological balance (*Mahé et al., 2021*). Studies have shown that imidacloprid is applied as a seed coating (*Simon-Delso et al., 2015*) and accumulates in the plant (*Dively and Kamel, 2012*; *Rocha et al., 2022*), exposing non-target bees to imidacloprid residues during pollen and nectar collection (*Tong et al., 2018*). Even low levels of systemic imidacloprid cause sublethal effects in adult bees, including neurotoxicity (*Belzunces et al., 2012*), behavioral changes (*Karahan et al., 2015*; *Medrzycki et al., 2003*), reproductive impairment (*Chaimanee et al., 2016*), reduced immunity (*Pettis et al., 2013*), apoptosis and autophagy (*Carneiro et al., 2022*), and shortened lifespan (*Anderson and Harmon-Threatt, 2021*). A recent finding shows that imidacloprid induces oxidative stress in honey bees (*Balieira et al., 2018*). This oxidative stress is a result of the long-term synergistic effect between imidacloprid and its target receptors (nAChRs) (*Ihara et al., 2020*) in bees, which leads to an excessive influx of $Ca^{2+}$ into the bee brain (*Farooqui, 2013*), disrupting $Ca^{2+}$ homeostasis and ultimately causing mitochondrial dysfunction and increased mitochondrial ROS generation, exacerbating oxidative stress.

The development and health status of juvenile animals play a critical role in population structure and future growth (*Gill et al., 2012*). However, they are more susceptible to the toxic effects of pesticides than adults. Despite this, there is limited information on the toxic effects of imidacloprid on juvenile animals. Available reports mainly focus on the epistatic effects of toxicity, with limited investigation into the involved biochemical mechanisms. Imidacloprid is known to pose a potential health risk to children through dietary exposure (*Anderson and Harmon-Threatt, 2021*), and may lead to developmental abnormalities in amphibians (*Samojeden et al., 2022*) and fish (*Islam et al., 2019*). Furthermore, negative effects on pupal development in insects have been observed (*Jia and Li, 2023*). For instance, cat fleas (*Ctenocephalides felis*) larvae treated with imidacloprid become immobile, and their intestines show pulsatile movements for up to 1 hr, eventually dying after 2 hr (*Mehlhorn et al., 1999*). Additionally, imidacloprid exposure can affect the average weight gain per day in *Harmonia axyridis* (Coleoptera: Coccinellidae) larvae (*Vincent, 2000*). It has also been found that imidacloprid exposure affects various cellular aspects, such as mitochondria, energy, lipids, and the transcriptome in *Drosophila* (*Felipe et al., 2020*), and prolongs the larval pupation duration and affects forewing development in the butterfly *Pieris brassicae* (*Whitehorn et al., 2018*).

In bees, larvae play a crucial role in the growth and development of colonies. However, the broader toxic effects of imidacloprid on bee larvae remain to be investigated. Limited reports suggest that developing bee larvae are exposed to imidacloprid through contact with contaminated hives or ingestion of contaminated food (*Böhme et al., 2018*). However, their survival, developmental rate, and body weight are not affected (*Dai et al., 2018*). Nevertheless, other studies have shown that imidacloprid

can lead to impaired olfactory learning behavior (*Peng and Yang, 2016*) and other behavioral abnormalities (*Wu et al., 2017*). The current understanding is that honey bee larvae are more tolerant to imidacloprid than adults, as they can survive treatment and develop into adults (*Dai et al., 2018*). However, they may experience developmental delays, such as delayed plumage (*Woyciechowski and Moroń, 2009*; *Wu et al., 2011*), and reduced rates of capping, pupation, and eclosion (*Yang et al., 2012*). A recent study using next-generation sequencing has indicated that sublethal imidacloprid treatment during the larval stage causes changes in gene expression in larvae, pupae, and adults, suggesting a persistent sublethal impact on honey bee development (*Chen et al., 2021*). Furthermore, a latest study has demonstrated that imidacloprid induces differential gene expression in bumble bee larvae, resulting in unique expression patterns related to various biological processes such as starvation response, cuticle genes, neural development, and cell growth (*Martín-Blázquez et al., 2023*).

Although progress has been made in understanding the toxic effects of imidacloprid on honey bee larvae, the molecular basis of this toxicity remains poorly understood. Further research is needed to uncover the mechanisms and pathways involved in the developmental toxicity of imidacloprid in honey bee larvae. In our pilot study, we observed a significant developmental delay in bee larvae exposed to imidacloprid; However, the molecular and biochemical mechanisms underlying this delay have not been systematically investigated. Therefore, this study aims to comprehensively evaluate the toxicity of imidacloprid on honey bee larvae at multiple levels, including histology, neurotoxicity, molting regulation, oxidative stress, detoxification, nutrition, and energy metabolism. Our study provides first-hand evidence of the developmental delay caused by imidacloprid in bee larvae. Because these mechanisms involved are relatively conserved in animals, our findings have broader implications for understanding and assessing the potential risk of continuous exposure to imidacloprid during animal development.

## Results

### Imidacloprid causes bee larval developmental retardation

The 96 hr observation period showed no significant difference in larval survival between the four groups exposed to imidacloprid and the control group. However, as the exposure concentration and duration increased, here was a significant decrease in larval developmental progress, weight, and width (*Figure 1A–E*). Further analysis was conducted on the most adverse exposure level of 377 ppb, which is the highest reported environmental residue concentration detected in bee products, such as beeswax (*Kapoor et al., 2014*). It is important to note that while the tested doses of the 377 ppb was found in bee products, it is likely that the bees themselves were exposed to even higher doses. Out result demonstrates that exposure to 377 ppb imidacloprid did not have a significant impact on survival (*Figure 1C* and *Figure 1—source data 1*), feeding (*Figure 1F* and *Figure 1—source data 4*, *Figure 1G*, and *Figure 1—source data 5*), and the time required for larvae to enter the pupal stage successfully (*Figure 1H* and *Figure 1—source data 6*). However, it did lead to slow developmental progress, a lower initial rate of successful pupation (*Figure 1I* and *Figure 1—source data 7*), and lower growth index (*Figure 1J* and *Figure 1—source data 8*). Larval growth index gradually decreased from the fourth instar, and then stabilized by the sixth instar. At the fourth, fifth, and sixth instars, the growth index of the imidacloprid-treated groups was significantly lower than that of the control group by an average of 1.35%, 4.49%, and 2.76%, respectively ($p<0.05$) (*Figure 1—source data 8*). In addition, imidacloprid consistently resulted in reduced larval width (*Figure 1D* and *Figure 1—source data 2*) and weight (*Figure 1E* and *Figure 1—source data 3*) up to the pupal stage (*Figure 1B*). Comparing the 377 ppb imidacloprid-exposed larvae to the control, the mean body weight and body width were significantly lower by 34.38% and 17.15%, respectively ($p<0.05$).

### Imdacloprid neurotoxicity in bee larvae

Under imidacloprid environmental stress at 377 ppb, acetylcholinesterase (AChE) activity, which is closely related to the disruption of neurotransmission, was significantly inhibited in larvae (*Figure 2A* and *Figure 2—source data 1*) and showed a significant average reduction of 48% ($p<0.05$) compared to the control group. Neither acetylcholine receptor (nAChR) alpha 1 (*Alph1*) nor acetylcholinesterase 2 (*Ace2*) gene expression was significantly different from the control. In contrast, acetylcholine

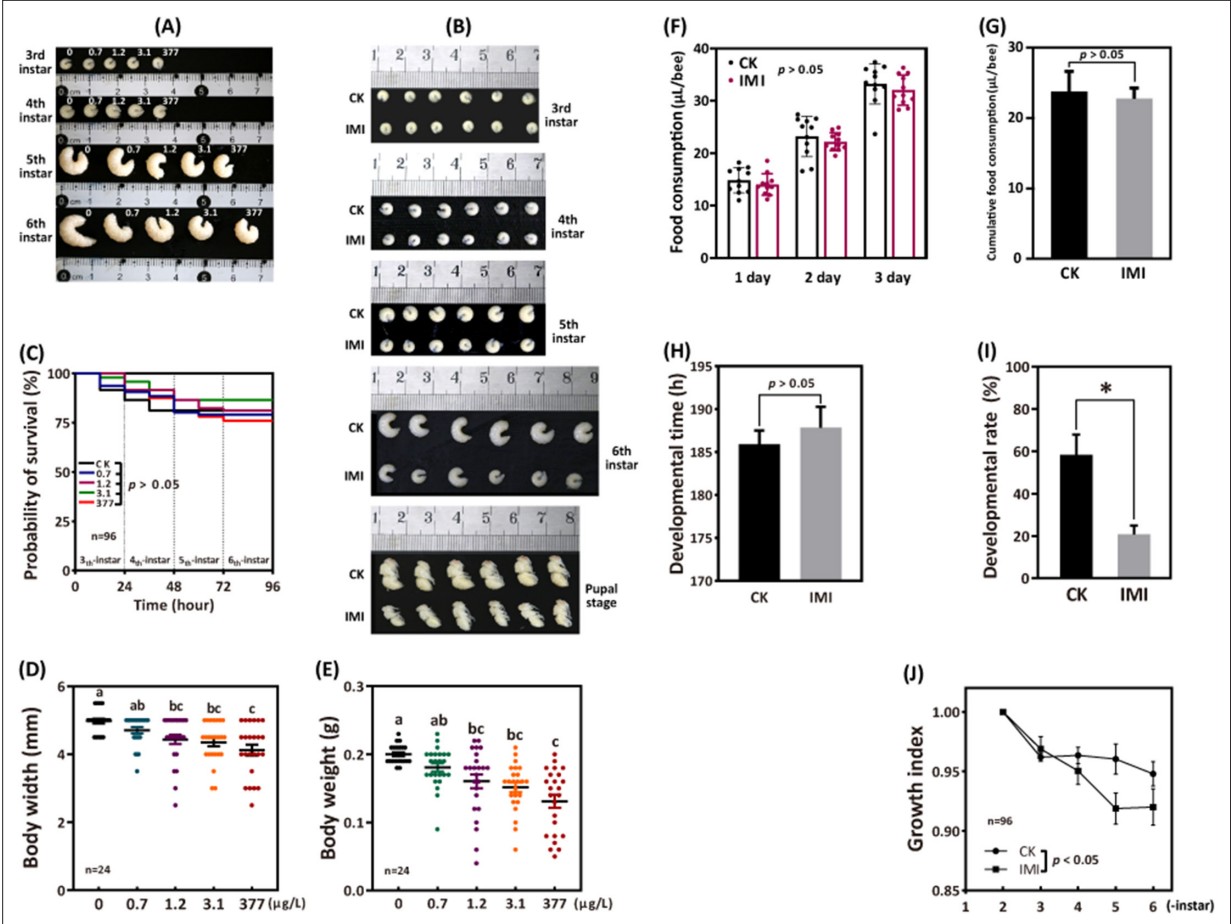

**Figure 1.** Imidacloprid causes larval developmental retardation in *A. mellifera*. Larvae were orally administered imidacloprid (IMI) or artificial food (CK) beginning at 3-day-old, followed by developmental monitoring for 96 hr. (**A**) Larval developmental phenotypes at 0.7, 1.2, 3.1, and 377 ppb environmental concentrations of imidacloprid exposure. (**B**) Larval developmental phenotype at 377 ppb imidacloprid exposure. (**C**) Effect of 0.7, 1.2, 3.1, and 377 ppb imidacloprid on larval survival. (**D** and **E**) Body width and weight of larvae exposed to 0.7, 1.2, 3.1, and 377 ppb imidacloprid for 96 hr. (**F** and **G**) Statistics for daily (**F**) and 3-day cumulative feeding (**G**) for larvae exposed to 377 ppb imidacloprid. (**H**) Larval development time from 3-day-old to pupal stage. (**I**) The number of larvae that successfully reached the pupal stage as a percentage of the total initial sample size. (**J**) The growth index of larvae. Statistical significance was set at *$p<0.05$ and **$p<0.01$.

The online version of this article includes the following source data for figure 1:

**Source data 1.** Effect of imidacloprid on the survival of *A. mellifera* larvae.

**Source data 2.** Effect of imidacloprid on the body width of *A. mellifera* larvae.

**Source data 3.** Effect of imidacloprid on the body weight of *A. mellifera* larvae.

**Source data 4.** Effect of imidacloprid on the food consumption of *A. mellifera* larvae.

**Source data 5.** Effect of imidacloprid on the cumlative food consumption of *A. mellifera* larvae.

**Source data 6.** Effect of imidacloprid on the developmental time of *A. mellifera* larvae.

**Source data 7.** Effect of imidacloprid on the developmental rate of *A. mellifera* larvae.

**Source data 8.** Effect of imidacloprid on the growth index of *A. mellifera* larvae.

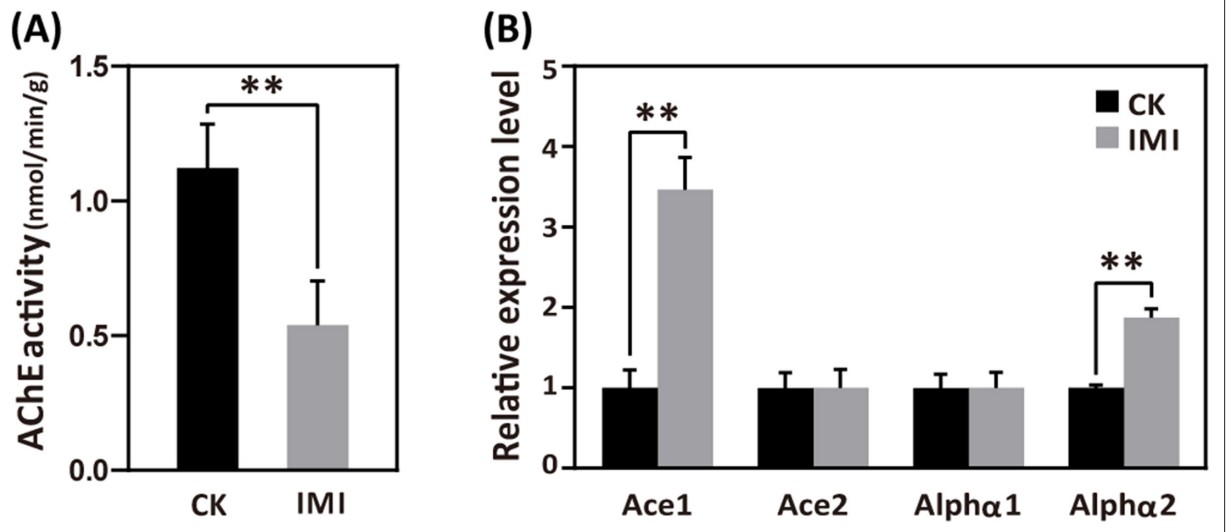

**Figure 2.** Imidacloprid neurotoxicity in *A. mellifera* larvae. (**A**) Acetylcholinesterase (AChE) activity and (**B**) nerve conduction-related gene expression analysis in larvae exposed to 377 ppb imidacloprid for 72 hr or a control group. IMI is the larvae exposed to imidacloprid for 72 hr. CK is the artificial food control group. Statistical significance was set at *p<0.05 and **p<0.01.

The online version of this article includes the following source data for figure 2:

**Source data 1.** The effect of imidacloprid on neurotoxicity in *A. mellifera* larvae such as the activity of AChE.

**Source data 2.** The effect of imidacloprid on neurotoxicity in *A. mellifera* larvae, including the expression of genes related to nerve conduction.

receptor alpha 2 (*Alph2*) and acetylcholinesterase 1 (*Ace1*) gene expression was significantly higher than that in the control group, with an average of 1.87 and 3.46 times higher (*Figure 2B* and *Figure 2—source data 2*), respectively (p<0.05).

## Imidacloprid disrupts the homeostasis of developmental regulation in larvae

In imidacloprid-exposed larvae, the developmental regulatory gene juvenile hormone acid methyl transferase (*JHAMT*) and the transcription factor broad complex (*Br-c*) were downregulated, while vitellogenin (*Vg*) gene expression was upregulated by sevenfold in comparison to control groups (p<0.05) (*Figure 3A* and *Figure 3—source data 1*). In parallel, imidacloprid exposure decreased the major regulator of insect metamorphosis 20-hydroxyecdysone (20E) titers but had no effect on preventing premature metamorphosis juvenile hormone (JH-3) titers (*Figure 3B* and *Figure 3—source data 2*).

## Imidacloprid upregulates larval detoxification genes expression

According to the qRT-PCR analysis, the expression levels of cytochrome P450 monooxygenase family members such as CYPq1, CYPq2, CYP450, CYP6AS14, CYP4G11, and CYP306A1, except for CYPq3, were observed to be significantly upregulated by several fold in imidacloprid-induced developmental retardation larvae. Among them, CYPq1 showed an abnormal 94-fold increase (p<0.05) (*Figure 4* and *Figure 4—source data 1*).

## Imidacloprid toxicity causes oxidative stress and induces antioxidant defense in larvae

Larvae exposed to imidacloprid exhibited a significant increase in reactive oxygen species (ROS) levels (*Figure 5A* and *Figure 5—source data 1*) and the activities of the antioxidants catalase (CAT), superoxide dismutase (SOD), and glutathione (GSH) (*Figure 5B* and *Figure 5—source data 2*), as well as upregulation of the antioxidant genes GPX and Trx expression (*Figure 5C* and *Figure 5—source data 3*), resulting in a significant increase in total antioxidant capacity (T-AOC) (*Figure 5B* and *Figure 5—source data 4*) compared to the control group. Under these conditions, malondialdehyde (MDA)

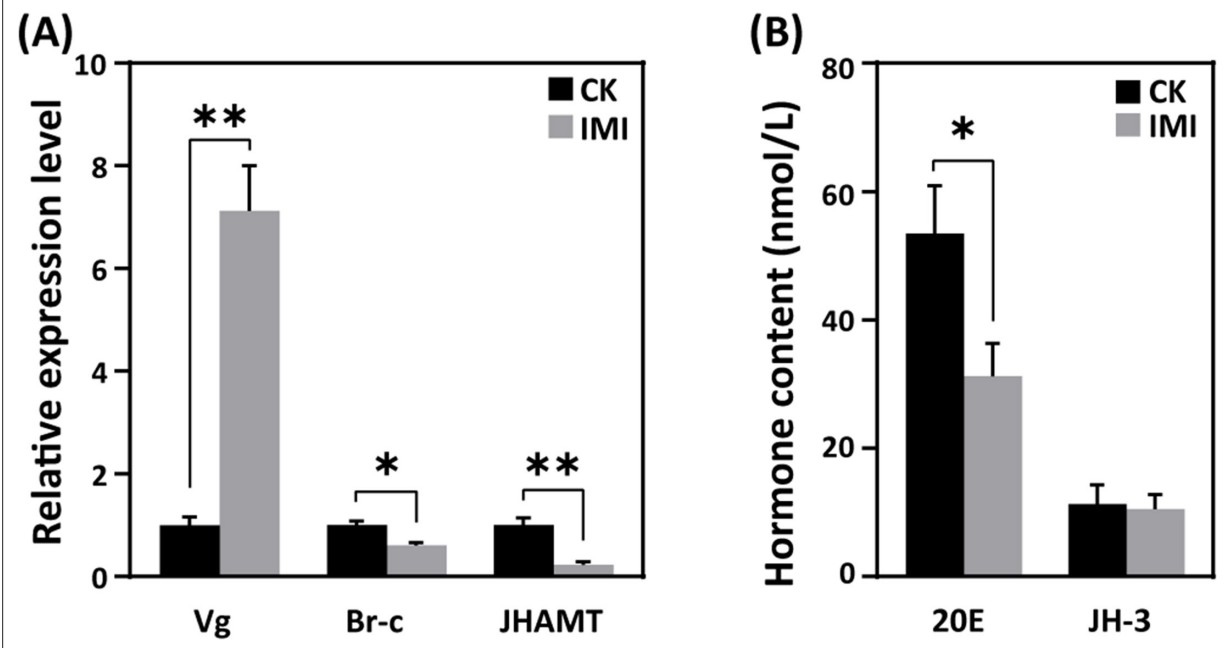

**Figure 3.** Effects of imidacloprid on the homeostasis of developmental regulation in *A. mellifera* larvae. (**A**) Relative gene expression and (**B**) hormone levels of developmental regulatory-related genes in larvae exposed to 377 ppb imidacloprid for 72 hr or a control group. Statistical significance was set at *p<0.05 and **p<0.01.

The online version of this article includes the following source data for figure 3:

**Source data 1.** Effect of imidacloprid on the expression of the developmental regulatory genes JHAMT, Br-c and Vg in *A. mellifera* larvae.

**Source data 2.** Effect of imidacloprid on the developmental regulatory hormones 20E and JH-3 in *A. mellifera* larvae.

levels, used as a marker of lipid peroxidation, were observed to significant increase of 58% (p<0.05), while protein carbonylation (PCO) damage indicator levels remained unchanged (*Figure 5A* and *Figure 5—source data 5*).

## Imidacloprid induces gut apoptosis and tissue damage in larvae

Hematoxylin-eosin (HE) staining revealed a significant reduction in the number of cells in the muscular layer of the larval gut exposed to imidacloprid with clear apoptotic characteristics compared to the standard feeding control groups (*Figure 5D*, *Figure 5—source data 4*, and *Figure 5—source data 5*). This was evidenced by increased nuclear staining and compacted chromatin, as well as disrupted cell arrangement in the basal layer of the gut, increased nuclear staining of basal lamina cells, and apoptotic signals. The integrity of the peritrophic membrane structure of the gut was better in the control group, while imidacloprid-exposed larvae were still in the formation stage, resulting in the accumulation of undigested food residues in their guts (*Figure 5D* and *Figure 5—source data 5*).

## Imidacloprid inhibits larval carbohydrate catabolism, proteolysis, and amino acid transporter gene expression

Exposure to imidacloprid did not significantly affect the daily food intake of developmentally delayed larvae compared to controls (*Figure 1F and G*, *Figure 1—source data 4* and *Figure 1—source data 5*). However, imidacloprid toxicity suppressed the expression of nutrient catabolism genes, including alpha-amylase (*α-Am*) and alpha-glucosidase (*α-Glu*), which are involved in carbohydrate catabolism (*Figure 6A* and *Figure 6—source data 1*), carboxypeptidase (*CPs*), and aminopeptidase (*APs*) involved in proteolysis (*Figure 6B* and *Figure 6—source data 2*), and glycine tRNA ligase (*Aats-Gly*) and tyrosine tRNA ligase (*Aats-Tyr*) involved in amino acid transport (*Figure 6C* and *Figure 6—source data 3*). Carbohydrate catabolism and amino acid transport were significantly suppressed by over 50% (p<0.05).

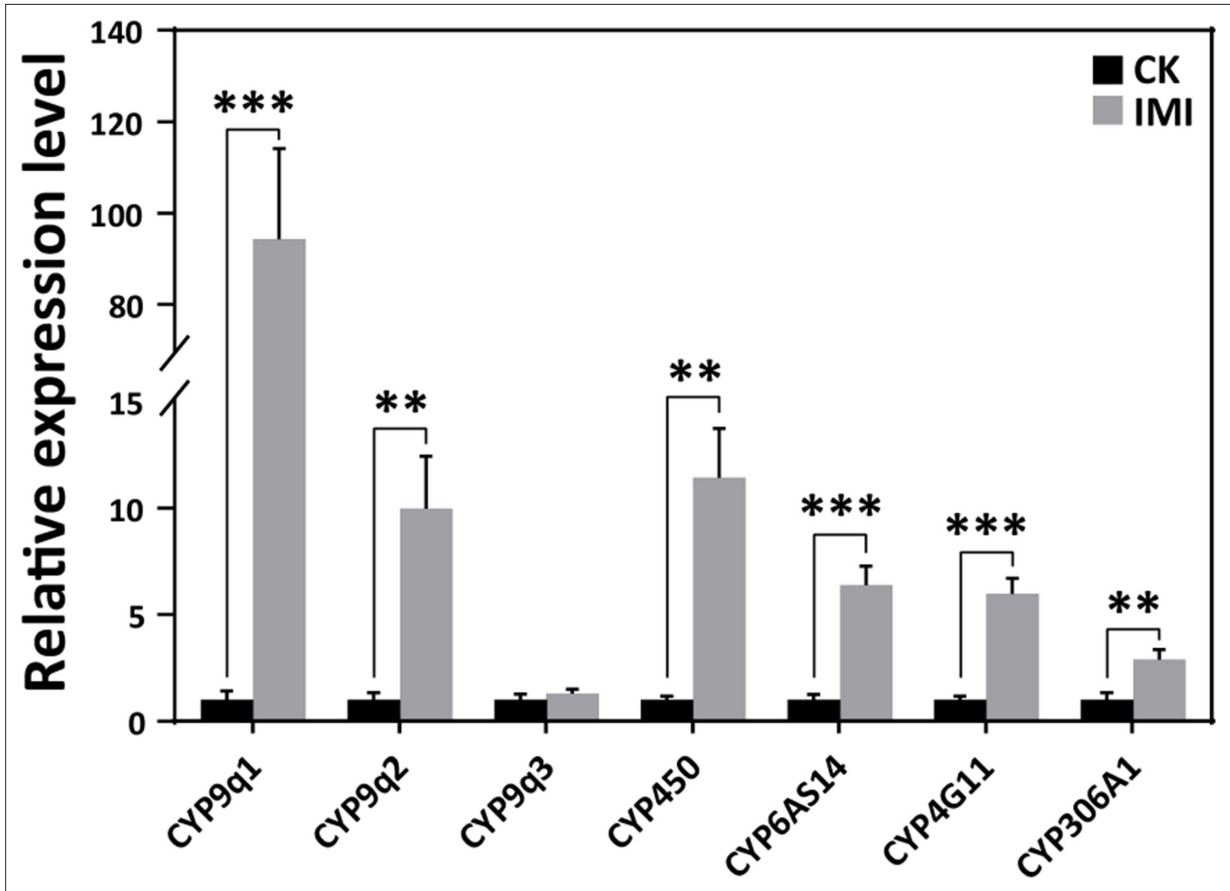

**Figure 4.** Imidacloprid induced a detoxification response in *A. mellifera* larvae. Relative expression levels of cytochrome P450 monooxygenase family genes in larvae exposed to imidacloprid for 72 hr. IMI is the larvae exposed to imidacloprid for 72 hr. CK is the artificial food control group. Statistical significance was set at *p<0.05, **p<0.01, and ***p<0.001.

The online version of this article includes the following source data for figure 4:

**Source data 1.** Effect of imidacloprid on the expression of detoxification genes in *A. mellifera* larvae.

## Imidacloprid causes restriction of larval protein synthesis and energy metabolic dysfunction

RT-qPCR analysis revealed that imidacloprid exposure reduced the expression of critical genes involved in energy metabolism, including cytochrome *c* oxidase copper chaperone (*COX17*) and NADH dehydrogenase (ubiquinone) 1β subcomplex subunit 7 (*NDUFB7*), in developmentally delayed larvae (*Figure 6D* and *Figure 6—source data 4*). Imidacloprid exposure also inhibited the expression of genes related to glycolysis and energy production, such as glyceraldehyde-3-phosphate dehydrogenase (*Gapdh*) and glucosamine-6-phosphate isomerase (*Oscillin*) (*Figure 6E* and *Figure 6—source data 5*). Both had expression levels below 50% of the control group (p<0.05). In addition, the expression of transcription initiation factors related to protein synthesis, including translation initiation factor 3 subunit M (*Tango7*), translation initiation factor 3 subunit B (*eIF3-S9*), and translation initiation factor 3 subunit A (*eIF3-S10*), was significantly suppressed by imidacloprid toxicity (*Figure 6F* and *Figure 6—source data 6*). Finally, the total levels of ATP, glycogen, and protein, which are closely related to energy production and reserves, were significantly lower in developmentally delayed larvae than in controls (p<0.05) (*Figure 6G–I*, *Figure 6—source data 7*, *Figure 6—source data 8*, and *Figure 6—source data 9*).

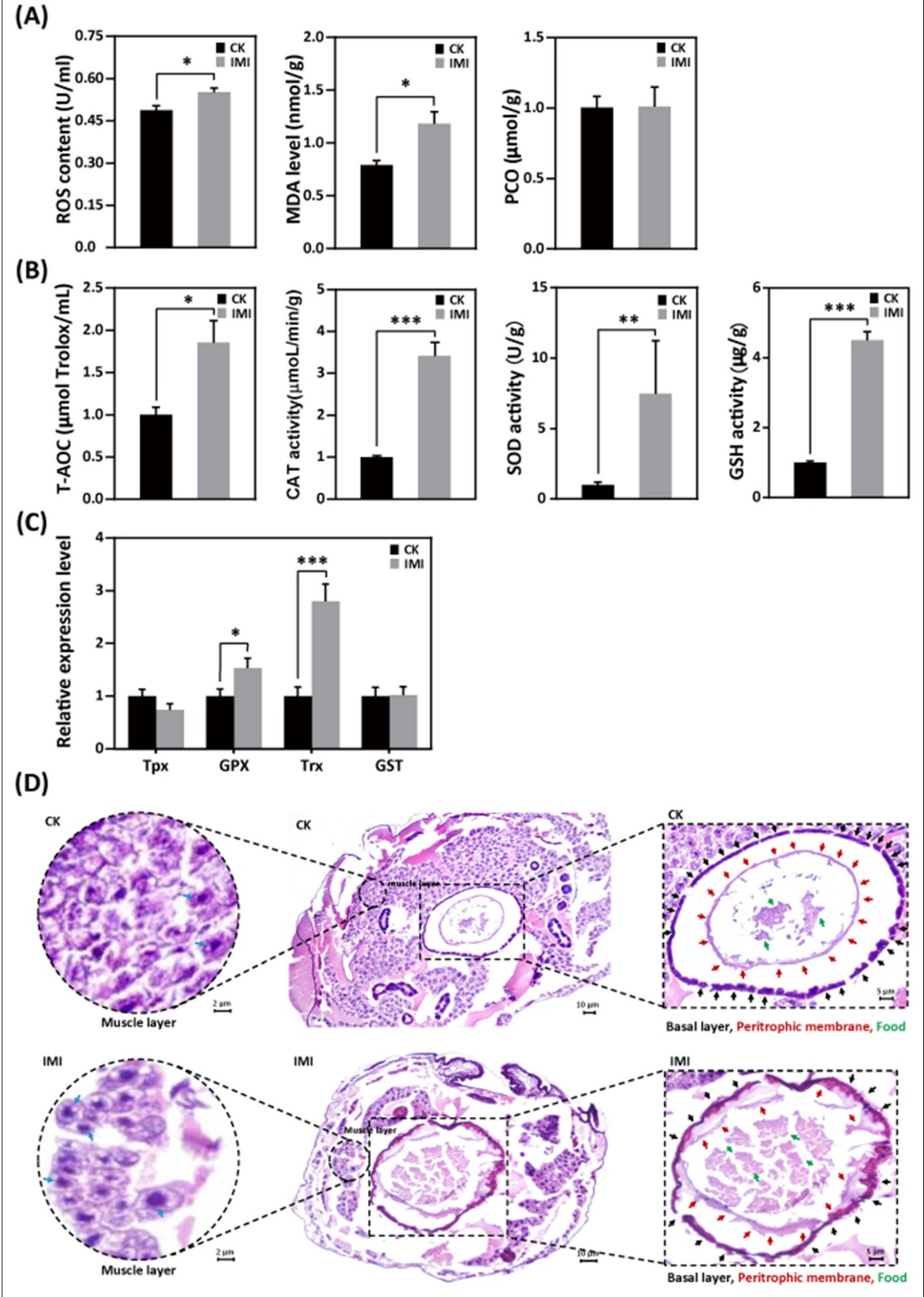

**Figure 5.** Imidacloprid exposure causes oxidative stress, gut apoptosis, and tissue structural damage while inducing antioxidant defenses in *A. mellifera* larvae. (**A**) Analysis of oxidative stress and damage in larvae exposed to imidacloprid for 72 hr. Reactive oxygen species (ROS) levels indicate the degree of oxidative stress. Malondialdehyde (MDA) is a marker for lipid peroxidation. Protein carbonylation (PCO) levels reflect the extent of protein carbonylation damage. (**B**) Activity of antioxidants in larvae exposed to imidacloprid for 72 hr. (**C**) The relative expression levels of antioxidant genes in

*Figure 5 continued on next page*

*Figure 5 continued*

larvae exposed to imidacloprid for 72 hr. (**D**) Histological sections of the larval gut stained with hematoxylin-eosin (HE). The circled inserts in the figure show magnified views of the muscle layer. The blue arrow indicates the cell nucleus of the muscle layer. The boxed inset in the figure is a magnified view of the basal layer (black arrow), peritrophic membrane (red arrow), and food residues (green arrow). IMI is the larvae exposed to imidacloprid for 72 hr. CK is the artificial food control group. Statistical significance was set at *p<0.05, **p<0.01, and ***p<0.001.

The online version of this article includes the following source data for figure 5:

**Source data 1.** Effect of imidacloprid on the ROS levels in *A. mellifera* larvae.

**Source data 2.** Effect of imidacloprid on the activities of the antioxidants CAT, SOD and GSH in *A. mellifera* larvae.

**Source data 3.** Effect of imidacloprid on the expression of antioxidant genes GPX and Trx in *A. mellifera* larvae.

**Source data 4.** Effect of imidacloprid on the total antioxidant capacity (T-AOC) in *A. mellifera* larvae.

**Source data 5.** Effect of imidacloprid on the oxidative damage in *A. mellifera* larvae.

## Correlation between developmental phenotypic changes and molecular features caused by imidacloprid toxicity

To better understand the mechanism of imidacloprid-induced larval developmental retardation, we performed a comprehensive correlation analysis between epigenetic growth traits and molecular characteristics (*Figure 7* and *Figure 7—source data 1*). The results showed that in larvae with imidacloprid-induced developmental retardation, body weight, width, growth index, and developmental rate were positively correlated with energy metabolism and reserves, nutrient catabolism, protein synthesis, and developmental regulation. Conversely, negative correlations were observed between these growth traits and parameters associated with detoxification and antioxidant defense processes that require additional energy expenditure. Moreover, oxidative stress and damage, detoxification, and antioxidant defense all exhibited significant positive correlations (*Figure 7* and *Figure 7—source data 1*).

# Discussion

## Toxic effects and the molecular basis of environmental concentrations of imidacloprid on bee larvae

Previous research has shown that bee larvae are more susceptible to the adverse effects of pesticides than adult bees (*Tomé et al., 2020*). In this study, larvae were exposed to environmental concentrations of imidacloprid, resulting in sublethal effects such as decreased body weight, width, developmental rate, and growth index (*Figure 1*). Our results support the notion that pesticides can affect larval development (*Dai et al., 2018*; *Zhu et al., 2014*). However, current research has primarily focused on characterizing the effects of imidacloprid on larval phenotype and behavior, and the molecular mechanisms underlying its ecotoxicological effects remain poorly understood.

To obtain molecular evidence of imidacloprid toxicity to bee larvae, we first examined the expression of acetylcholine receptors and AChE activity. The results showed that imidacloprid exposure increased the expression of the *Alph2* gene and inhibited AChE activity (*Figure 2* and *Figure 2— source data 2*). Normally, acetylcholine stimulates nerve impulses by binding to the Alph receptor, and AChE degrades acetylcholine to terminate nerve conduction. However, imidacloprid competitively binds to AChE receptors, inhibits AChE activity, and causes continuous excitation of nerve conduction, ultimately leading to paralysis and death (*Katić et al., 2021*; *Shan et al., 2020*). Therefore, in the present study, imidacloprid induced the expression of the acetylcholine receptor *Alph2* and inhibited the expression of AChE activity and its encoding gene *Ace1*, which inevitably led to more than sustained excitation of larval nerve conduction, suggesting that imidacloprid is neurotoxic to honeybee larvae. In addition, we also noted that apart from *Alph2* and *Ace1*, imidacloprid did not have any effect on the expression of the additional acetylcholine receptor *Alph1* and the AChE *Ace2*, which may be related to the fact that sometimes changes in gene expression do not necessarily correlate with changes in receptor function and degradation. But more importantly, this result may indicate that the nerve impulses of the larvae are regulated in many ways, and that the toxicity of the imidacloprid is actually insufficient to affect the entire neural conduction of the larvae. A more

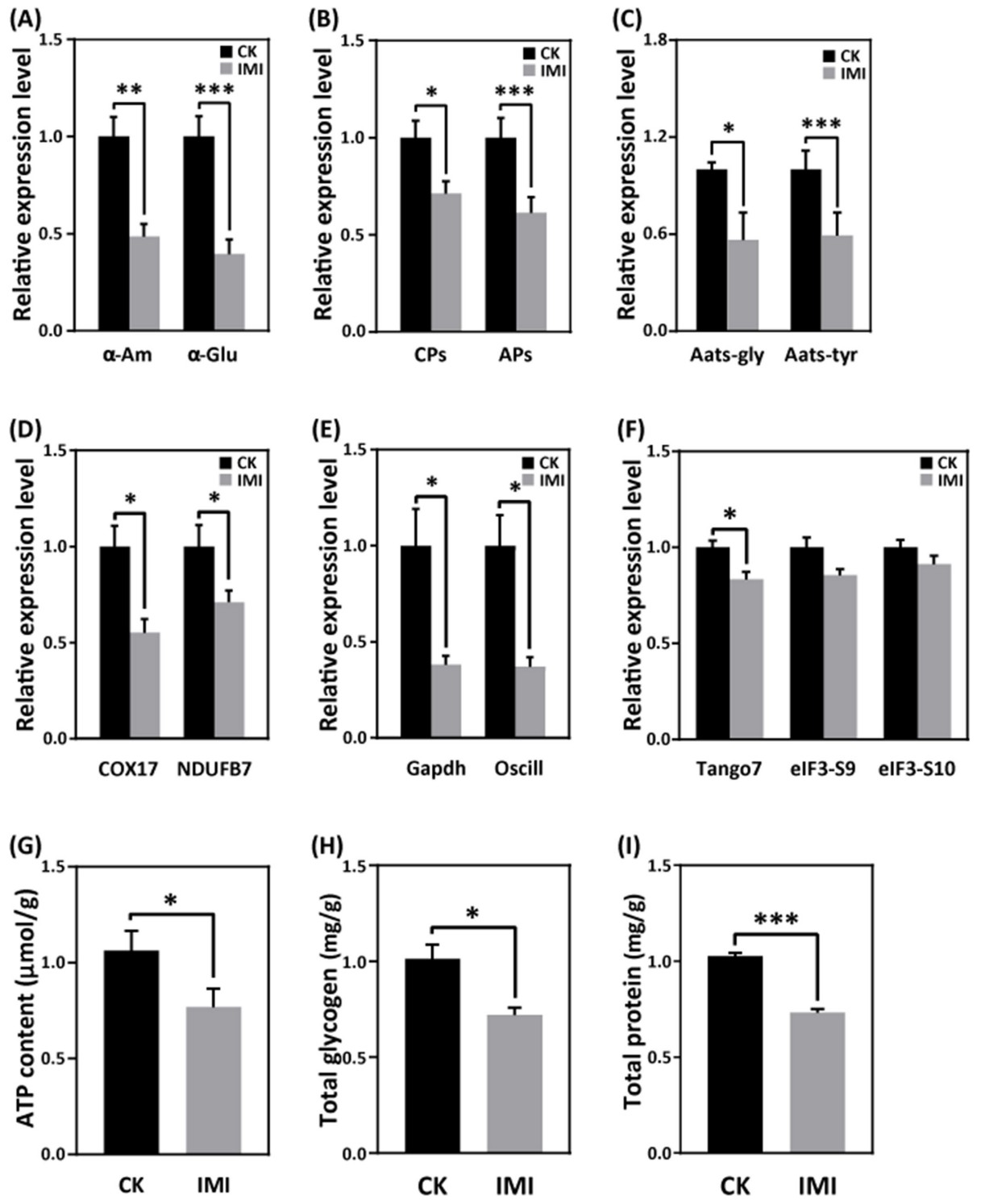

**Figure 6.** Imidacloprid toxicity inhibited the expression of genes involved in nutrient catabolism, protein synthesis, and energy metabolism and reserves. (**A**) Carbohydrate catabolism. (**B**) Proteolysis. (**C**) Amino acid transport. (**D**) Mitochondrial oxidative phosphorylation. (**E**) Glycolysis. (**F**) Protein synthesis. (**G**) The total contents of ATP, glycogen (**H**), and protein (**I**). IMI is the larvae exposed to imidacloprid for 72 hr. CK is the artificial food control group. Statistical significance was set at *p<0.05, **p<0.01, and ***p<0.001.

The online version of this article includes the following source data for figure 6:

**Source data 1.** Effects of imidacloprid on α-Am and α-Glu, genes involved in carbohydrate metabolism in *A. mellifera* larvae.

*Figure 6 continued on next page*

*Figure 6 continued*

**Source data 2.** Effects of imidacloprid on CPs and APs, genes involved in proteolysis in *A. mellifera* larvae.

**Source data 3.** Effects of imidacloprid on Aats-Gly and Aats-Tyr, genes involved in amino acid transport in *A. mellifera* larvae.

**Source data 4.** Effects of imidacloprid on COX17 and NDUFB7, genes involved in energy metabolism in *A. mellifera* larvae.

**Source data 5.** Effects of imidacloprid on Gapdh and Oscillin, genes involved in glycolysis and energy production in *A. mellifera* larvae.

**Source data 6.** Effects of imidacloprid on Tango7, eIF3-S9, and eIF3-S10, genes involved in protein synthesis in *A. mellifera* larvae.

**Source data 7.** Effects of imidacloprid on total ATP levels involved in energy production and reserves in *A. mellifera* larvae.

**Source data 8.** Effects of imidacloprid on total glycogen contents involved in energy production and reserves in *A. mellifera* larvae.

**Source data 9.** Effects of imidacloprid on total protein levels involved in energy production and reserves in *A. mellifera* larvae.

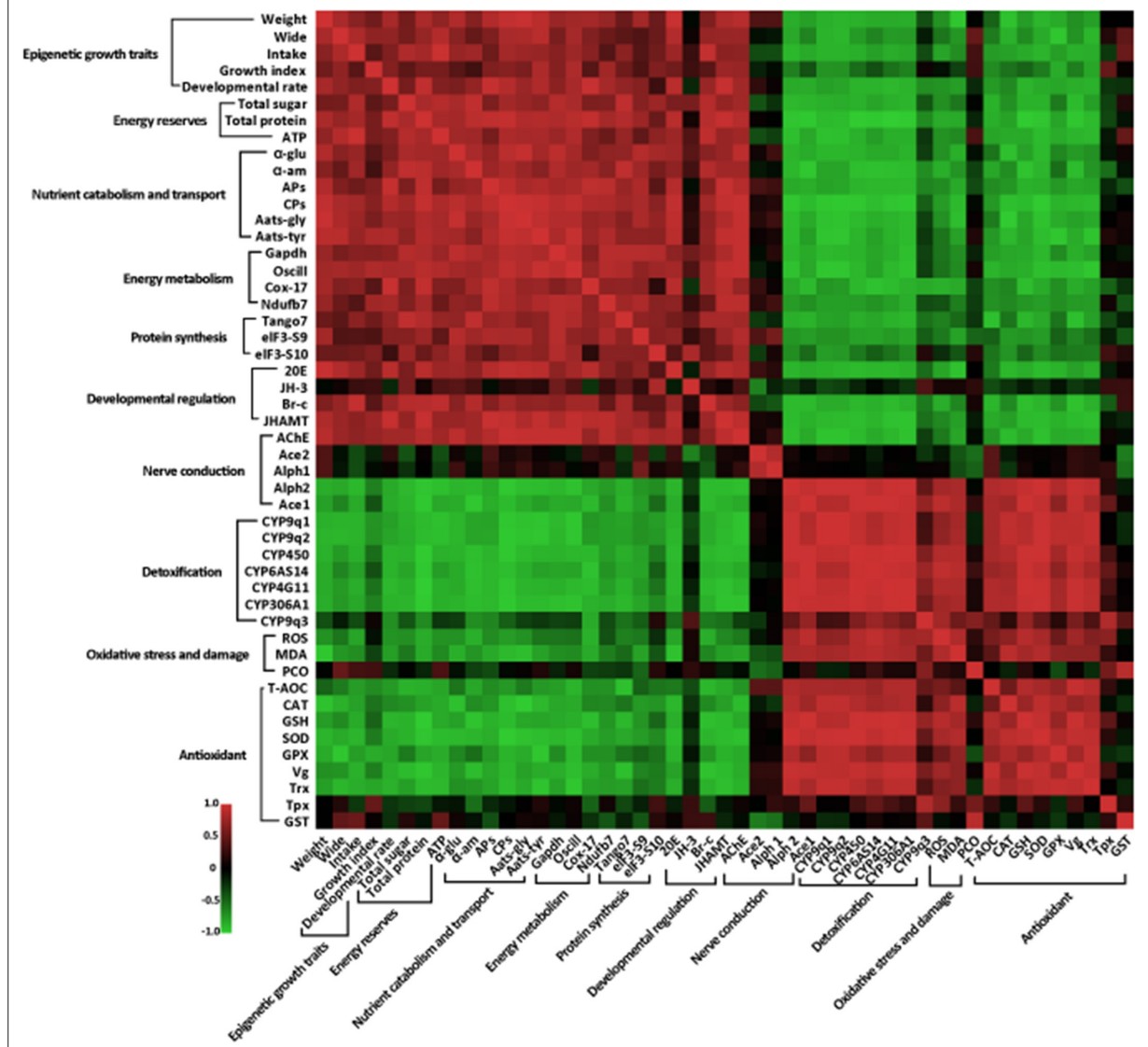

**Figure 7.** Correlation between phenotypes and molecular characteristics in imidacloprid-exposed larvae with developmental retardation. Data were normalized using GraphPad software, followed by a Pearson correlation coefficient calculation using SPSS, and a heatmap was generated.

The online version of this article includes the following source data for figure 7:

**Source data 1.** All data used for correlation analysis between developmental retardation phenotypes and molecular characteristics.

comprehensive investigation of most of the nAChRs targets in bee larvae will help to provide new insights into how pesticides cause neurotoxicity.

The growth and development of juvenile animals depends on the digestion, absorption, and utilization of food nutrients. Therefore, we next investigated whether imidacloprid affects the digestion and utilization of dietary by larvae. The results indicate that exposure to imidacloprid suppressed the hydrolysis genes *CPs* and *APs* and the amino acid transport pathway genes *Aats-Gly* and *Aats-Tyr* in larvae (*Figure 6B–C*, *Figure 6—source data 2*, and *Figure 6—source data 3*). Since these genes are responsible for cleaving dietary proteins into absorbable peptides and amino acids and transporting them for metabolism (*Sharma et al., 2019*), inhibiting the expression of these genes would severely affect larval digestion, resulting in protein deficiency and impaired metabolic activity. In addition, imidacloprid also affected the expression of *α-Am* and *α-Glu* genes (*Figure 6A* and *Figure 6—source data 1*), which are involved in carbohydrate digestion, indicating that imidacloprid also affects the digestion of dietary carbohydrates. Indeed, histological observations also confirmed that a large amount of undigested food remained in the intestine of larvae exposed to imidacloprid (*Figure 5D* and *Figure 5—source data 5*). In conclusion, imidacloprid inhibits the digestion and utilization of dietary proteins and carbohydrates by larvae, which is extremely detrimental to their growth and development.

In addition to nutrition, energy metabolism is critical for the maintenance of all life activities. In this study, imidacloprid exposure inhibited the expression of genes related to mitochondrial oxidative phosphorylation (*COX17*, *NDUFB7*) and its alternative glycolytic pathway (*Gapdh*, *Oscillin*). Mitochondrial oxidative phosphorylation is the central pathway for ATP generation during energy metabolism (*Tian et al., 2020a*), while glycolysis is an alternative mechanism for ATP synthesis (*Jiang et al., 2020*). Since both biochemical processes are essential for energy production, their inhibition resulted in reduced ATP levels in bee larvae (*Figure 6G* and *Figure 6—source data 7*). These results strongly suggest that imidacloprid induces energy metabolism dysfunction leading to insufficient energy production in larvae. Since larval energy production is insufficient, this is likely to lead to a depletion of energy reserves. Indeed, further analysis revealed a significant decrease in glycogen and protein energy reserves in larvae exposed to imidacloprid (*Figure 6H–I*, *Figure 6—source data 8*, and *Figure 6—source data 9*). These findings are consistent with previous reports that exposure to pesticides such as endosulfan, chlorpyrifos, insecticides, and fipronil can lead to a reduction in energy reserves in organisms (*Bouayad et al., 2012*; *Dutra et al., 2009*; *Radwan et al., 2008*; *Rambabu and Rao, 1994*; *Ribeiro et al., 2001*). In conclusion, our results suggest that imidacloprid induces energy metabolism dysfunction and affects energy production and reserves in bee larvae.

Insects rely primarily on cytochrome P450 detoxification to resist toxic compounds (*Li et al., 2007*), and the level of CYP450 expression indicates the severity of pesticide toxicity. While adult worker bees have shown a positive response to imidacloprid (*Mao et al., 2009*), it is uncertain whether bee larvae can respond to pesticide toxicity. In this study, we found that seven cytochrome P450 family transcripts, including CYPq1, CYPq2, CYP450, CYP6AS14, CYP4G11, and CYP306A1, were significantly upregulated in bee larvae, with CYP9q1 being 94-fold higher than the control group (*Figure 4* and *Figure 4—source data 1*). These results are consistent with previous findings that the CYP450 family genes CYP6AS3, CYP6AS4, and CYP9S1 were significantly upregulated in adult worker bees exposed to quercetin, coumaphos, and fluvalinate (*Mao et al., 2011*) and that CYP9Q1, CYP9Q2, and CYP9Q3 help bees detoxify fluvalinate and coumaphos (*Gregorc et al., 2018*). This suggests that the P450 detoxification system is already established in bee larvae and they respond positively by activating the detoxification function of CYP450 upon exposure to imidacloprid.

Oxidative stress is the excessive production of ROS in aerobic organisms (*Felton, 1995*). Pesticides, including coumaphos (*Gregorc et al., 2018*), fipronil (*Paris et al., 2017*), organophosphorus, pyrethroid, organochlorinated (*Chakrabarti et al., 2015*), and chlorpyrifos (*Shafiq-ur-Rehman and Waliullah, 2012*), as well as the herbicide such as paraquat (*Li-Byarlay et al., 2016*), have been shown to cause oxidative stress in bees. In this study, exposure to imidacloprid resulted in increased levels of ROS and MDA in larvae (*Figure 5A* and *Figure 5—source data 1*), indicating severe oxidative stress and lipid damage. Organisms have evolved antioxidant defense mechanisms to scavenge excessive ROS and resist oxidative stress, including increasing levels of antioxidants such as CAT, SOD, GSH, TrxR, and GSH (*Dong et al., 2013*; *Tian et al., 2020b*). In this regard, we have recently confirmed this in worker bees (*Li et al., 2022*), which is also well reflected in the present study. We observed a

significant increase in the activities of antioxidant enzymes SOD and CAT, gene expression of Trx and Gpx, and GSH levels in larvae after exposure to imidacloprid (*Figure 5B–C*, *Figure 5—source data 2* and *Figure 5—source data 3*). These results show that the antioxidant system in developing larvae is established and used to resist oxidative stress caused by imidacloprid, similar to that in adult worker bees. Notably, compared to our recent study in adult bees (*Li et al., 2022*), the larvae in this study exhibited more robust antioxidant activity when exposed to higher concentrations of imidacloprid toxicity than worker bees. However, the underlying mechanism requires further investigation. The underlying mechanisms involved require further study.

In addition to the many toxic damages to bee larvae, we are interested in whether the low impact of imidacloprid on larval growth rate is problematic for adult bees or colony health, because there have been a small number of reports that pesticide damage to bee larvae has subsequent negative effects on adult bee traits such as morphology, foraging efficiency, and olfactory-related behaviors (*Tan et al., 2015*; *Wu et al., 2011*). In this study, we attempted to investigate the behavior and physiological traits of adult bees developed from imidacloprid-stressed larvae in this study to investigate the impacts of developing larval health on adult individuals and the colony, but unfortunately, due to technical constraints, we were unable to artificially breed them into adults under laboratory conditions. Although the data on this topic is deficient, especially at the molecular level, it deserves much attention.

## Mechanisms of imidacloprid-induced developmental retardation in bee larvae

Impaired juvenile development can have a significant impact on the growth and sustainability of entire populations. Deviations in juvenile development due to environmental stress have recently received considerable attention. *Drosophila melanogaster* exposed to heat stress exhibit developmental delays due to the failure of early embryos to fully inhibit the synthesis of non-heat shock proteins (*Bergh and Arking, 1984*). Similarly, exposure to nutrient and heavy metal stress can cause delayed or arrested development in *Caenorhabditis elegans* due to altered gene expression and disruption of developmental signaling pathways (*Carranza-García and Navarro, 2020*; *Rashid et al., 2021*). In the present study, we also observed typical developmental delays in honeybee larvae exposed to imidacloprid, but the underlying mechanisms remain unclear.

Several mechanisms may be responsible for the delayed development of honey bee larvae, including endocrine disruption, altered gene expression of metabolic pathways, and increased energy expenditure due to detoxification mechanisms. Nonetheless, no evidence has been reported in detail to date. Insect development is regulated by the hormones JH and 20E, which control morphological remodeling events during molting. 20E promotes pupation (*Yuan et al., 2020*), while JH maintains juvenile characteristics and prevents metamorphosis by antagonizing the effects of 20E (*Luo et al., 2021*). Multiple studies show that neonicotinoid insecticides disrupt honey bee endocrine systems (*Christen et al., 2018*; *Cook, 2019*). This suggests that imidacloprid may affect molt-related endocrine hormones. Nevertheless, this phenomenon has not been reported in honeybees yet. Therefore, we first focused on the hormonal regulation of molting development and found that imidacloprid inhibits the expression of JHAMT (*Figure 3* and *Figure 3—source data 1*), but did not affect JH titer levels (*Figure 3B* and *Figure 3—source data 2*). JHAMT is an enzyme that converts JH acids or inactive precursors of JHs to active JHs at the final step of the JH biosynthesis pathway in insects, which is a positive regulator of JH synthesis (*Liu et al., 2018*). Therefore, our results suggest that the antagonistic effect of JH still restricts the developmental process, thereby maintaining larval morphology. A more important finding was that imidacloprid toxicity resulted in reduced ecdysteroid 20E titer and Br-c expression (*Figure 3*, *Figure 3—source data 1*, and *Figure 3—source data 2*). As insects develop, 20E helps the larvae shed their skin and turn into pupae by activating response factors like Br-c (*Deng et al., 2012*). As a result, the lower levels of ecdysteroid 20E and Br-c found in this study suggest that the honeybee larval normal moulting development process was slowed down. Therefore, our results suggest that imidacloprid may reduce 20E titer and inhibit *Br-c* expression, thereby blocking molting and causing larval developmental delay. Notably, we also observed that a decrease in 20E levels was accompanied by a decrease in AChE activity (*Figure 2A* and *Figure 2—source data 1*), which is known to cause delayed pupation (*He et al., 2012*) and developmental arrest (*Desneux et al., 2007*) in insect larvae. Therefore, our results suggest that imidacloprid neurotoxicity

may cause developmental delay in bee larvae by inhibiting the molting hormone 20E. This conclusion is supported by a correlation analysis showing that decreases in 20E titer, *Br-c* expression, and AChE activity were positively correlated with decreased developmental rates and growth index, weight, and width growth (*Figure 7* and *Figure 7—source data 1*). In conclusion, imidacloprid neurotoxicity inhibits the 20E and *Br-c* genes, thereby blocking molting, which may be an important cause of delayed larval development in bee larvae. However, we cannot rule out the possibility that the decline in the growth index may be due to other factors, such as oxidative stress impairing mitochondria, dysregulated neuro-endocrine axis caused by imidacloprid targeting neurons, poor nutrient absorption, impaired movement, etc., as animal growth and development are collectively regulated by numerous physiological, biochemical, and genetic factors.

Animals employ selective foraging as a survival strategy to avoid ingesting toxic foods that may cause adverse physiological effects (*Berenbaum and Johnson, 2015*). Although bees cannot actively reject toxins innately, postingestive malaise allows them to learn to avoid ingesting toxic-containing foods with adverse physiological effects (*Hurst, 2014*). Given this, we speculate that it is likely that the bee larvae in this study reduced their intake of toxic imidacloprid-containing foods, making it difficult to meet their nutritional and energy needs, which may have resulted in delayed development. However, monitoring of daily food intake did not show a significant difference between larvae exposed to imidacloprid and the control group (*Figure 1F and G*, *Figure 1—source data 4*, and *Figure 1—source data 5*), indicating that larval food intake was not reduced and larvae did not initiate avoidance behavior toward toxic food. Therefore, food intake was not a critical factor in the delayed larval development caused by imidacloprid.

Nutrition and energy are critical for the growth and development of juvenile animals. Exposure to heavy metals and neonicotinoid pesticides has been shown to affect digestion, energy reserves, production, and utilization in animals (*Jiang et al., 2020*; *Li et al., 2021*). However, our study found no correlation between food intake and larval developmental delay, so we focused on food utilization for nutrition and energy. Interestingly, we found severe pathological changes in the gut tissues of developmentally delayed larvae exposed to imidacloprid (*Figure 5D* and *Figure 5—source data 4*). Imidacloprid caused a significant reduction in the tightness of the cell arrangement in the basal layer of the gut, and a drastic decrease in the number of cells. This inevitably impacts the larval digestion and absorption of food nutrients. Indeed, the peritrophic membrane was incompletely formed, and undigested food residues were found in the gut of imidacloprid-exposed larvae compared to controls. In addition, qRT-PCR evidence showed a significant decrease in digestion and catabolism of dietary proteins and carbohydrates, amino acid transport, oxidative phosphorylation, and glycolysis (*Figure 6D–F*, *Figure 6—source data 4*, *Figure 6—source data 5*, and *Figure 6—source data 6*). Biochemical assays showed that imidacloprid stress also decreased ATP levels and total glycogen and protein content, resulting in inadequate energy reserves and increased energy expenditure (*Figure 6G–I*, *Figure 6—source data 7*, *Figure 6—source data 8*, and *Figure 6—source data 9*). Furthermore, correlation analysis provided some support for a positive relationship between imidacloprid-induced impaired catabolism and dietary nutrient utilization and larval developmental delay (*Figure 7* and *Figure 7—source data 1*). In conclusion, these findings clearly suggest that imidacloprid exposure affects the digestion and utilization of nutrients and energy in larvae, resulting in insufficient nutrients and energy for growth and development, which may be another important factor contributing to imidacloprid-induced larval developmental delay.

In addition, mitochondria are the main producers of ATP for cellular metabolism, accounting for approximately 90% of the total. However, mitochondria are also involved in the generation of ROS. Excessive accumulation of ROS in mitochondria leads to oxidative stress, which in turn damages mitochondria and further increases ROS levels, creating a vicious cycle (*Boovarahan and Kurian, 2018*). In the present study, it was found that imidacloprid exposure led to increased ROS and MDA levels in larvae (*Figure 5A* and *Figure 5—source data 1*), indicating that imidacloprid induced severe oxidative stress and lipid damage, which may damage mitochondria and in turn affect mitochondrial ATP production, resulting in insufficient energy supply for larval development. This factor may also be an important explanation for the larval developmental delay caused by imidacloprid.

The dynamic energy budget theory proposes that environmental stress can significantly affect the energy balance of organisms, leading to increased energy expenditure (*Kooijman, 2009*). To cope with stress, animals may deplete their energy reserves, such as glycogen and protein (*Bouayad et al.,*

*2012*; *Matsukura et al., 2008*). In this study, we found that larvae with delayed development caused by imidacloprid had lower energy reserves (ATP, total protein, and total glycogen). Furthermore, this decrease in energy reserves was negatively correlated with larval antioxidant and detoxification (*Figure 7* and *Figure 7—source data 1*). It is commonly acknowledged that antioxidant defense and detoxification for stress resistance is an energy-consuming process. Therefore, our results indirectly imply that imidacloprid-induced reduction in energy reserves is likely associated with the energy-consuming activities of CYP450 detoxification and antioxidant defense. In nature, to meet the high energy cost demands of environmental stress, animals often have to allocate most of their food intake to defense (*Guedes, 2006*), and this redistribution of energy resources will inevitably lead to an insufficient energy supply for growth and development (*Beyers et al., 1999*). However, on this point, it remains to be supported in the future by overexpression of P450 and antioxidant enzymes, combined with complete metabolome and transcriptome analysis of the overall expression of other P450s, antioxidant genes to obtain direct evidence that depletion of larval antioxidants and detoxification leads to a decrease in the growth index.

It is worth noting that our investigation into the causes of imidacloprid-induced developmental delay in honey bee larvae has provided limited answers. Future comparative analyses of different tissues and durations of exposure, using genetic manipulation and histochemical methods, would improve our understanding of the effects of sublethal doses of imidacloprid on larval development, growth, and survival.

## Conclusion

Imidacloprid toxicity caused growth and developmental delay in *A. mellifera* larvae. The toxic mechanism may include imidacloprid disrupting the regulatory balance of molt development, resulting in restricted development. In addition, the gut damage caused by imidacloprid restricts the metabolism and utilization of food nutrients and energy in the larvae. Third, the additional energy consumed by

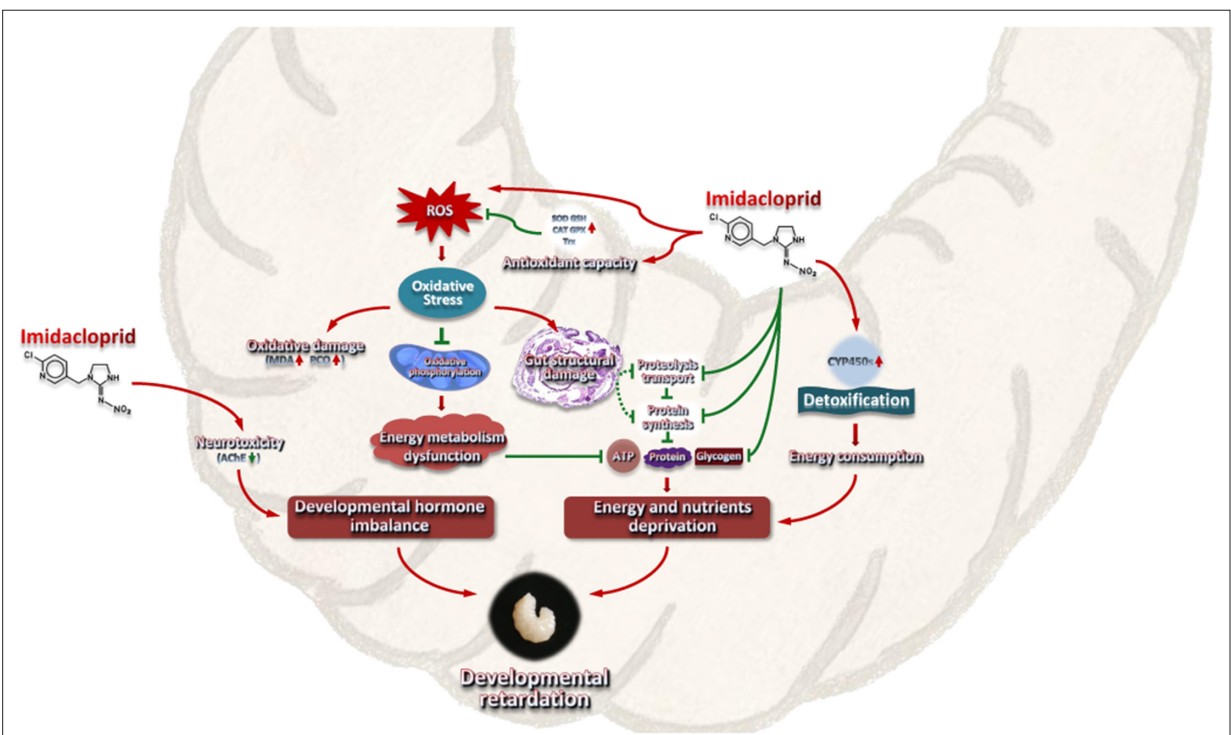

**Figure 8.** A model interpreting the ecotoxicological effects of imidacloprid on *A. mellifera* larvae and the mechanisms of imidacloprid-induced developmental retardation. Imidacloprid toxicity caused growth and developmental delay in *A. mellifera* larvae. It disrupts the regulatory balance of molt development, resulting in restricted development. In addition, the damage caused by imidacloprid restricts the metabolism and utilization of food nutrients and energy in larvae. The additional energy consumed by larval P450 detoxification and antioxidant defenses further reduces the supply of nutrients and energy supply for growth and development. The arrows indicate induction or promotion, the straight lines indicate inhibition, and the green dotted lines indicate speculative inhibition.

larval P450 detoxification and antioxidant defenses further reduces the supply of nutrients and energy for growth and development (*Figure 8*). This study is the first multi-level investigation of the toxic effects of imidacloprid on *A. mellifera* larvae. These findings have broader implications for understanding and assessing the threat of harmful pesticides to animal larvae.

## Materials and methods

### Larval rearing

The protocol of the current research was approved by the Animal Bioethics Committee of the Chongqing Normal University, China (Protocol Number: 201001). The chosen colonies were healthy and not exposed to pathogens or pesticides. Two-day-old larvae from the same frames of the same hive were individually transferred to sterile 24-well cell culture plates. The plates contained standard food, which included royal jelly, glucose, fructose, water, and yeast extract (*Aupinel et al., 2005*). The larvae were cultured in dark conditions at 35°C and 96% RH, with daily feeding volumes adjusted according to their age (20 µL for 3-day-old larvae, 30 µL for 4-day-old larvae, 40 µL for 5-day-old larvae, and 50 µL for 6-day-old larvae). Any remaining food was recorded and removed.

### Experimental design

This study employed four different concentrations of imidacloprid, which were residual concentrations found in honey (0.7 ppb) (*Chauzat et al., 2009*), bees (1.2 ppb) (*Chauzat et al., 2011*), pollen (3.1 ppb) (*Mullin et al., 2010*), and beeswax (377 ppb) (*Kapoor et al., 2014*). The experiment was conducted with three treatment groups: control group (CK), solvent control group (CKac), and imidacloprid-treated group (IMI), each with three replicates. Each replicate contained 32 larvaes that were distributed in standard 24-well cell culture plates (1 bee well$^{-1}$). The CK group consisted of larvaes fed acetone-royal jelly glucose-fructose-yeast extract-distilled water (0.00377%−50%−6%−6%−1%−36.99623%, vol/wt/wt/wt/wt/vol). The IMI group consisted of the 2-day-old larvae that were orally exposed to a control solution containing imidacloprid (99% active ingredient, Hubei Norna Technology Co. Ltd, Hubei, China) at 0.7, 1.2, 3.1, and 377 ppb, respectively, after which the larvae were fed a standard food from 3-day-old. The CKac group served as a solvent control and contained only acetone at a concentration of 0.00377%. A total of 312 larvae were used for survival statistics (96 bees), body weight and width (96 bees), and gene expression, enzyme activity, histochemical content, and histopathological analysis. Food was administered daily according to age (20 µL for 3-day-old larvae, 30 µL for 4-day-old larvae, 40 µL for 5-day-old larvae, and 50 µL for 6-day-old larvae). Dead individuals and food remain were recorded daily, and incubation conditions were 35°C and 96% RH. Surviving larvae were collected 72 hr after consumption of the test solution and stored at −80°C for further analysis, including evaluations of gene expression, enzyme activity, and chemical content evaluations. Furthermore, HE-stained sections were used to evaluate any imidacloprid-induced tissue damage and apoptosis.

### Developmental phenotypic traits

Larval individuals were randomly selected daily from the CK and IMI groups to measure body weight and width. Food residues, larval mortality, developmental progression, and the amount of time spent in the larval developmental stage (from the first imidacloprid exposure to before pupa) were recorded daily to calculate food consumption, survival rates, developmental rates, and development times. The growth index was calculated for larvae exposed to imidacloprid from the 3-day-old to the 6-day-old using the equations described by *Zhang et al., 1993*. The experiment was conducted in three biological replicates, each containing 24 randomly selected larvae (24-well cell culture plate, 1 bee well$^{-1}$). The 2-day-old larvae were administered imidacloprid orally (after which they were maintained on standard chow), and then the developmental progress of the larvae was monitored daily for 12 days starting at 4-day-old, and the developmental time and developmental rate were counted. Sample data that died abnormally or failed to enter the pupal stage during this period were excluded. Developmental time was calculated as the length of time from 4-day-old until the larvae entered the pupal stage after oral administration of imidacloprid to 2-day-old larvae. Developmental rate was defined as the percentage of the number of larvae that developed into pupae out of 24 two-day-old larvae that received oral doses of imidacloprid.

The developmental rate was measured as the number of larvae successfully reaching the pupal stage as a percentage of the total initial sample size.

## RNA extraction, cDNA synthesis, and real-time quantitative PCR analysis

Total RNA was extracted from larvae using TRIzol reagent (Thermo Fisher Scientific) and RNAex Pro Reagent (Accurate Biology, China) as instructed by the manufacturer. The RNA quality was evaluated by gel electrophoresis and absorbance determination using a NanoDrop 2000 spectrophotometer (Thermo Scientific, New York, USA). Genomic DNA was removed, and cDNA was synthesized using a reverse transcription kit (Evo M-MLV RT Mix Kit with gDNA Clean for qPCR, Accurate Biology, China) according to the manufacturer's instructions. Briefly, 1 µg of RNA reacted with the 5× gDNA Clean Reaction Mix provided in the kit at 42°C for 5 min to remove residual genomic DNA to ensure the accuracy of the quantitative results. After the reaction, 4 µL of 5× Evo M-MLV RT Reaction Mix and 4 µL of RNase-free water were added to the reaction tube for 20 µL. The mixture was incubated at 37°C for 15 min and 85°C for 5 s to generate cDNA. Then, quantitative RT-PCR was performed with a Bio-Rad CFX96 Real-time PCR Detection System (Bio-Rad, Hercules, CA, USA) according to standard protocols and cycling conditions. RP49 (ribosomal protein 49, Accession no. AF441189) (**Lourenço et al., 2008**) was used as the endogenous control. The primers for target genes and RP49 are listed in **Supplementary file 1**. qPCR system (20 µL): EvaGreen Express 2× qPCR MasterMix (10 µL), each forward and reverse primer (0.4 µL, 0.2 µmol L$^{-1}$), cDNA (1 µL, 100 ng µL$^{-1}$), ddH$_2$O (8.2 µL). The thermal cycling program was as follows: 95°C for 30 s, followed by 40 cycles of 95°C for 5 s and 60°C for 30 s. The qPCR products were sequenced by Shanghai Biological Engineering Technology Services Co., Ltd. to confirm whether the amplified fragments were the target sequences. Relative mRNA expression levels of target genes were calculated using the $2^{-\Delta\Delta Ct}$ method.

## Histochemical measurement

Juvenile hormone 3 (JH-III), 20-hydroxyecdysone (20E) titers, and ROS levels in larvae were determined using ELISA (enzyme-linked immunosorbent assay) kits (Jining, Shanghai, China) according to the manufacturer's instructions. Briefly, the larva was homogenized on ice in 5 volumes (M:V) of phosphate-buffered saline (PBS) buffer, and the supernatant was collected after centrifugation at 12,000 × $g$ for 20 min. Subsequently, 50 µL of reference standard solution or 10 µL of sample supernatant plus 40 µL of dilution were added to the microtiter wells and incubated at 37°C for 60 min with 100 µL of HRP-labeled detection antibody. Each well was washed five times with 400 µL of PBS buffer containing 0.05% Tween 20, pat-dried, and left to stand at room temperature for 5 min. TMB chromogen solution A (50 µL) and solution B (50 µL) were added to each well, gently mixed, and incubated at 37°C for 15 min in the dark. Then, 50 µL of H$_2$SO$_4$ (4 mol L$^{-1}$) was added to each well to terminate the reaction. Finally, the OD value was measured at 450 nm with a NanoDrop 2000 spectrophotometer (Thermo Scientific, New York, USA), and the content of the detected substance was calculated from a calibration curve constructed by reference standards.

To prepare samples for analysis of PCO, AChE, CAT, MDA, T-AOC, SOD, and GSH, larvae was weighed and homogenized on ice in 5 volumes (M:V) of PBS buffer, then centrifuged at 12,000 × $g$ for 15 min to collect the supernatant. Subsequently, PCO content was measured using the 2,4-dinitrophenyl-hydrazine (DNPH) method with slight modifications. The above-prepared supernatant was mixed with streptomycin sulfate at a 9:1 volume (V:M), kept at room temperature for 10 min, and then centrifuged at 12,000 × $g$ for 15 min. Then, 80 µL of supernatant was mixed with 160 µL of 10 mM DNPH in 2 M HCl and reacted at 37°C for 30 min. Sample controls were also prepared by adding an equal volume (500 µL) of 2 M HCl. Then, 200 µL of 20% TCA was added, mixed thoroughly, and centrifuged at 4°C for 10 min at 12,000 × $g$, and the precipitate was collected. Add 400 µL of ethanol:ethyl acetate mixture (1:1), vortex and mix, centrifuge at 4°C 12,000 × $g$ for 10 min, and collect the precipitate to remove unbound DNPH. This DNPH removal step was repeated once. The pellet was mixed with 400 µL of 6 M guanidine hydrochloride, kept at room temperature for 15 min, and then centrifuged at 4°C 12,000 × $g$ for 10 min to collect the supernatant. The carbonyl and protein contents were determined by reading the absorbance at $\lambda$ = 370 nm and 280 nm with a SpectraMax 190 spectrophotometer (Molecular Devices, CA, USA) for each sample against an appropriate blank. Finally, the carbonyl content per unit of protein was calculated based on the protein content.

AChE activity was measured by the Ellman method as described by *Orhan et al., 2007*, with slight modifications. Briefly, 20 µL of the prepared sample supernatant was mixed with 160 µL of 0.1 mM PBS buffer (pH 8.0) and 10 µL of an AChE (0.28 U mL$^{-1}$) and incubated for 15 min at room temperature. Then, 10 µL of ATChI (acetylthiocholine iodide, 4 mg mL$^{-1}$) was added together with 20 µL of DTNB (5,5-dithiobis-2-nitrobenzoic acid, 1.2 mg mL$^{-1}$) and immediately mixed, and the absorbance was read at 412 nm. The results were expressed as activity percentages relative to the sample solvent negative control.

The CAT activity was determined using a kit (Beyotime, Shanghai, China) according to the manufacturer's instructions with slight modifications. Briefly, 10 µL of the prepared sample supernatant was mixed with 30 µL of Tris-HCl buffer (pH 7.0, 50 mmol L$^{-1}$), 10 µL of 250 mM $H_2O_2$ was added, and the mixture was immediately mixed and kept at 25°C for 5 min. Then, 450 µL of 1.8 mol L$^{-1}$ $H_2SO_4$ was added, and the mixture was vortexed to terminate the reaction. Subsequently, 10 µL of this termination reaction solution was mixed with 40 µL of Tris-HCl buffer (pH 7.0, 50 mmol L$^{-1}$). Then, 10 µL of this mixture was added to a 96-well plate mixed with 200 µL of color development working solution containing peroxidase and incubated for 15 min at 25°C. Finally, the absorbance was measured at 520 nm, and the CAT activity was calculated using a calibration curve of 5 mM $H_2O_2$.

T-AOC was measured using the T-AOC assay kit with the ABTS (2,2'-azino-bis(3-ethylbenzthiazoline-6-sulfonic acid)) method based on a previous report (*Chen et al., 2019*). Add 10 µL of the prepared sample supernatant to 200 µL of ABTS working solution. After 6 min of reaction, the absorbance was measured at 734 nm. Finally, the antioxidant activity was displayed by Trolox equivalents antioxidant capacity as mmol Trolox equivalents g$^{-1}$ extract. The ABTS working solution was prepared as follows: 400 µL of potassium persulfate was added to 400 µL of ABTS to generate an ABTS stock solution, which was then incubated for 16 hr at room temperature in the dark. Before use, the stock solution was diluted with phosphate buffer, and the absorbance at 734 nm was adjusted to 0.70±0.05.

Additionally, MDA, SOD, and GSH levels were detected using ELISA kits (Beyotime, Shanghai, China) according to our recent study (*Li et al., 2022*).

## Histopathological analysis

Microsectioning and HE staining was performed according to standard histological protocols (*Khan et al., 2010*) with some modifications. Briefly, entire fresh larvae were fixed in 10% neutral-buffered formalin solution for 12 hr at 4°C after 72 hr of imidacloprid exposure. After isopropyl alcohol dehydration, samples were embedded in paraffin and sectioned. Three-micron-thick sections were stained with a Hematoxylin and Eosin Staining Kit (Beyotime, Shanghai, China), observed with an Olympus BX51 microscope, and photographed with an Olympus digital camera.

## Statistical analysis

Data were obtained from three independent experiments performed in triplicate and are presented as the mean ± SEM. SPSS statistical package was used for statistical analysis using Student's t-test and one-way ANOVA followed by Tukey's multiple comparison test (*p<0.05, **p<0.01, and ***p<0.001 were considered statistically significant). GraphPad software was used for data normalization, Pearson correlation coefficient calculation was analyzed using SPSS, and a heatmap was generated.

## Acknowledgements

The authors would like to thank the other colleagues in our laboratory who contributed to this study. This study received financial support from the grants from the earmarked fund for China Agriculture Research System (No. CARS-44); the Science and Technology Project of the Chongqing Municipal Education Commission of China (No. KJZD-K202100502); the Natural Science Foundation Project of Chongqing of China (No. cstc2021jcyj-msxmX0422); and the Natural Science Foundation Project of the State Key Laboratory of Silkworm Genome Biology of China (No. SKLSGB-ORP202103).

# Additional information

## Funding

| Funder | Grant reference number | Author |
|---|---|---|
| China Agriculture Research System | CARS-44 | Zhi Li |
| Chongqing Municipal Education Commission | KJZD-K202100502 | Zhi Li |
| Natural Science Foundation Project of Chongqing of China | cstc2021jcyj-msxmX0422 | Zhi Li |
| Natural Science Foundation Project of the State Key Laboratory of Silkworm Genome Biology of China | SKLSGB-ORP202103 | Zhi Li |

The funders had no role in study design, data collection and interpretation, or the decision to submit the work for publication.

## Author contributions

Zhi Li, Conceptualization, Resources, Data curation, Formal analysis, Supervision, Funding acquisition, Validation, Investigation, Methodology, Writing – original draft, Project administration, Writing – review and editing; Yuedi Wang, Data curation, Software, Investigation, Methodology, Writing – original draft, Project administration; Qiqian Qin, Lanchun Chen, Data curation, Software, Validation, Investigation, Methodology; Xiaoqun Dang, Software, Formal analysis, Investigation, Methodology; Zhengang Ma, Data curation, Software, Investigation, Methodology; Zeyang Zhou, Conceptualization, Resources, Project administration

## Author ORCIDs

Zhi Li (iD) https://orcid.org/0000-0003-1611-4873

## Ethics

The protocol of the current research was approved by the Animal Bioethics Committee of the Chongqing Normal University, China (Protocol Number: 201001).

Joint Public Review: https://doi.org/10.7554/eLife.88772.4.sa1
Author Response https://doi.org/10.7554/eLife.88772.4.sa2

# Additional files

## Supplementary files

• Supplementary file 1. Primers used in real-time quantitative PCR.

• MDAR checklist

## Data availability

All data generated or analysed during this study are included in the manuscript and supporting file; source data files have been provided for figures.

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
