## [Editor Report · eLife assessment]

This investigation of the changes in gene expression and some of the physiological consequences of sublethal exposures to the neonicotinoid insecticide imidacloprid in honeybee larvae is **useful**, although numerous experiments were not considered based on technical issues. The methodological design leads to concerns and it is therefore not obvious that all conclusions are justified. The study adds to our understanding of how this insecticide impacts development and growth of honeybees, but the evidence supporting the major claims is **incomplete**.

---

## [Referee Report · Joint Public Review]

This study provides evidence on the ability of sublethal imidacloprid doses to affect growth and development of honeybee larva. While checking the effect of doses that do not impact survival or food intake, the authors found changes in the expression of genes related to energy metabolism, antioxidant response, and P450 metabolism. The authors also identified cell death in the alimentary canal, and disturbances in levels of ROS markers, molting hormones, weight, and growth ratio. The study strengths come from employing these different approaches to investigate the impacts of imidacloprid exposure. The study weaknesses come from the lack of a in depth investigation and drawing many conclusions solely from punctual gene expression, that are not representative of complete biological processes. Though relevant to understand the impacts on neonicotinoid contamination on insect pollinators, the study conclusions should be carefully weighted as they are often not fully substantiated. Follow up studies using in-depth investigation and more robust methodological design testing whether the impacts observed lead to post-metamorphosis effects and impacts in the colony would have a significant impact.

---

## [Author Response]

The following is the authors’ response to the previous reviews.

On behalf of all the authors, I'd like to thank you for your insightful comments and valuable suggestions, which fully reflect your high level of scientific thinking and point the direction of our research and help us and other future researchers in the field to more comprehensively study and interpret the toxic effects of imidacloprid on honey bee larvae and its potential mechanisms, as well as the mechanisms of larval resistance and adaptations to imidacloprid. We have addressed each of the questions and revised the manuscript point-by-point in response to your comments. Below are detailed point-by-point responses to each question.

**Public Review:**
This study provides evidence of the ability of sublethal imidacloprid doses to affect growth and development of honeybee larva. While checking the effect of doses that do not impact survival or food intake, the authors found changes in the expression of genes related to energy metabolism, antioxidant response, and metabolism of xenobiotics. The authors also identified cell death in the alimentary canal, and disturbances in levels of ROS markers, molting hormones, weight and growth ratio. The study strengths come from exploring different aspects and impacts of imidacloprid exposure on honeybee juvenile stages and for that it demonstrates potential for assessing the risks posed by pesticides. The study weaknesses come from the lack of in depth investigation and an incomplete methodological design. For instance, many of the study conclusions are based on RT-qPCR, which show only a partial snapshot of gene expression, which was performed at a single time point and using whole larvae. There is no understanding of how different organs/tissues might respond to exposure and how they change over time. That creates a problem to understand the mechanisms of damage caused by the pesticide in the situation studied here. There is no investigation of what happens after pupation. The authors show that the doses tested have no impact on survival, food consumption and time to pupation, and the growth index drops from ~0.96 to ~0.92 in exposed larvae, raising the question of its biological significance. The origin of ROS are not investigated, nor do the authors investigate if the larvae recover from the damage observed in the gut after pupation. That is important as it could affect the adult workers' health. One of the study's central claims is that the reduced growth index is due to the extra energy used to overexpress P450s and antioxidant enzymes, but that is based on RT-qPCR only. Other options are not well explored and whether the gut damage could be causing nutrient absorption problems, or the oxidative stress could be impairing mitochondrial energy production is not investigated. These alternatives may also affect the growth index. The authors also state that the honeybee larvae has 7 instars, which is an incorrect as Apis mellifera have 5 larval instars. It is not clear from methods which precise stage of larval development was used for gut preparations. That information is important because prior to pupation larvae defecate and undergo shedding of gut lining. That could profoundly affect some of the results in case gut preparations for microscopy were made close to this stage. A more in-depth investigation and more complete methodological design that investigates the mechanisms of damage and whether the exposures tested could affect adult bees may demonstrate the damage of low insecticide doses to a vital pollinator insect species.
**Recommendations for the authors:**
This study presents a useful investigation on changes in gene expression by real time PCR and some of the physiological consequences of sublethal exposures to the neonicotinoid insecticide imidacloprid in honeybee larvae. It offers preliminary evidence of imidacloprid impacts on the development of bee larvae by interfering with molting and metabolism. Whereas the study provides evidence that small doses of imidacloprid affect larval growth rate, there is no investigation on whether that could affect the overall colony health, and some of the results open the possibility that the larvae may overcome some of the impacts of the exposure. As the authors state, the doses tested show no impact on larvae survival, food consumption or time to pupation. The investigation and methodological design lack in depth to explain the findings and provide incomplete evidence to support the authors conclusions. The study would benefit from a more thorough mechanistic characterization to better sustain the findings and demonstrate their biological relevance.

Response: I would like to express, on behalf of all the authors, our sincere appreciation for your insightful and insightful comments and suggestions, which significantly enhanced the quality of the manuscript. Your incisive insights point the way for future research in the field of bee biology on the mechanisms underlying imidacloprid-induced delays in larval development.

In this study, we investigated the effects of imidacloprid on honey bee larval development, including macro and micro changes and possible causes. This is the first of its kind in the field of honeybee biology research. However, we found that the underlying mechanism is extremely complex. The effects of toxic substances on animals and their interactions with larval development are complex and far-reaching. They include oxidative stress and damage; disruption of nutrient metabolic homeostasis; inhibition of detoxification and immunity; adverse effects on the nervous, circulatory, and digestive systems; inflammation, disease, and even organ failure; and subsequent effects on physiological activities such as development, reproduction, and behavior, and even death. These toxic effects interact in complex ways with the development of young animals, with some effects directly or indirectly affecting development while others do not.

Addressing this complex mechanistic issue based solely on the results of this study is a formidable challenge, which leads to some limitations of our study as pointed out by the reviewers. Although our study is not comprehensive enough in terms of mechanistic analysis and does not fully elucidate the mechanism, we believe it is an important and valuable first step in this area.

In the future, we will follow the reviewers' suggestions and deliberately redesign the experiments to focus on further research on the issues they raised. These include examining the effects of larval developmental delay on adult and colony health, investigating the post-pupal situation, identifying the source of ROS, and determining whether the larval gut damage observed after pupalization recovers.

In accordance with the reviewers' comments and suggestions, we have revised the manuscript to improve its rigor and scientific quality. We sincerely ask the reviewers to understand and accept this modification from us!

Next is our response to each of the questions and valuable suggestions provided by reviewers:

**Recommendations For The Authors:**
1. The authors found a reduction in growth index and body mass, but document no impact on survival, food consumption or time to pupariation. How much exactly is the reduction in growth index? It seems to be from ~0.96 to ~0.93. Is this biologically relevant? Would that be enough to impact the colony health?

Response: Thank you for your comments. In this study, we observed a gradual decrease in larval growth index from day 4, which stabilized by day 6. At the 4th, 5th and 6th instars, the growth index of the imidacloprid-treated groups were significantly lower than those of the control group by an average of 1.35%, 4.49% and 2.76%, respectively (Figure 1, source data 8). Statistical analysis confirmed the significance of the difference in these results. We have incorporated the above description into the red text on lines 148-152 of the Results section. Regarding the reviewer's inquiry on colony health, including imidacloprid-induced delayed larval development and some reduction in growth index and body weight with no effect on survival, food consumption, or time develop to pupation, because we do not currently have the technical capabilities to culture larvae to adulthood in laboratory incubators, this has resulted in a failure to further investigate the effects of imidacloprid-induced delayed larval development on adult colony health. However, this is a very important scientific question for future colony health. We will design experiments to address this issue in a follow-up study.

1. The authors find that P450s can help in detoxifying mechanisms to mitigate imidacloprid impacts. That however is a well-known fact. What is new about this claim?

Response: The point at which the ability to detoxify toxic substances is acquired during early development varies widely among animals. Although many studies have reported that the detoxification function of P450s helps mitigate the effects of imidacloprid in adult honey bees, there is no conclusive evidence as to whether or not honey bee larvae have acquired this ability at early stages of development. This ability is critical to the defense and health of honey bee larvae. Therefore, it is incumbent upon this study to clarify this issue, which is important in explaining the effects of imidacloprid on honey bee larvae.

1. Some references are cited incorrectly. The first and last name are swapped, for instance Charles et al.

Response: Thank you very much for pointing out this error, which we have corrected. Please see lines 92 and 889 in our revised version.

1. I still encounter important methodological flaws. The authors acknowledge my previous suggestions but only address a small fraction of them. The most relevant points regarding the understanding of the mechanisms behind the delayed growth rate remain unexplored. The expression levels of other nAChRs target of imidacloprid in honeybees were not investigated. The expression analyses are still based on a single time point and using whole larvae, which only superficially explore the problem and may lead to misinterpretations. I do not understand the authors claim that a technological breakthrough is required to address these issues, when performing more PCRs and doing dissections should cover the matter.

Response: Thank you very much for your important comment. You point out several unexplored issues related to understanding the mechanisms behind delayed growth rates. For example, The most relevant points regarding the understanding of the mechanisms behind the delayed growth rate remain unexplored. The expression levels of other nAChRs target of imidacloprid in honeybees were not investigated. The expression analyses are still based on a single time point and using whole larvae. Please allow me to explain. Honeybees (Apis mellifera) have nine different α-subunits, Amelα1-9, and two β-subunits, Amelβ1-2. Amelα5, Amelα7, and Amelα8 are expressed in MB Kenyon cells and AL neurons, and the Amelβ2 subunit is present in Kenyon cells. Amelα2, Amelα3, and Amelα7-2 are expressed in the optic lobes. The aim of this study was to investigate whether imidacloprid induces larval neurotoxicity. Based on the above information, we selected the two most representative nAChRs (Alph1 and Alph2) for analysis. The results showed that exposure to imidacloprid increased the expression of the Alph2 gene and inhibited AChE activity, indicating that imidacloprid is neurotoxic to larvae. This result answered our question of whether imidacloprid induces neurotoxicity in larvae. Therefore, we did not further analyze the expression levels of other nAChRs. We believe that this does not affect the understanding of the mechanism behind the delayed growth rate and that it is not necessarily necessary to analyze all 11 nAChRs to find an answer. We sincerely hope that the reviewers will understand and agree with this.

Furthermore, regarding the expression analysis based on a single time point and whole larvae. In this study, 72 h after imidacloprid exposure Fig. 1J, 5 days of age was chosen for sampling because this is when imidacloprid has the greatest and most representative effect on larval development. Therefore, analyzing samples at this time point did not interfere with our exploration of the mechanisms by which imidacloprid causes larval developmental retardation. We used whole larvae rather than individual tissues for sample selection, which is a shortcoming for us. This was mainly due to technical challenges where we were unable to obtain pure single tissues through dissection. Nevertheless, we will make technical breakthroughs in the future so that we can sample and compare different tissues and developmental stages to obtain more comprehensive and accurate data. Thank you again for raising this important issue and for your valuable suggestions.

1. The authors could in many different ways explore what are the origin of ROS is. That is important to further develop their hypothesis on reduced energy levels.

Response: Thank you very much for your insightful comment and suggestion, it gives us great insight. Mitochondria are the main producers of ATP for cellular metabolism, accounting for approximately 90% of the total. However, mitochondria are also involved in the generation of reactive oxygen species (ROS). Excessive accumulation of ROS in mitochondria leads to oxidative stress, which in turn damages mitochondria and further increases ROS levels, creating a vicious cycle (Boovarahan and Kurian, 2018). In the present study, it was found that imidacloprid exposure led to increased ROS and MDA levels in larvae (Figure 5A and Figure 5-source data 14), indicating that imidacloprid induced severe oxidative stress and lipid damage, which may damage mitochondria and in turn affect mitochondrial ATP production, resulting in insufficient energy supply for larval development. This factor may also be an important explanation for the larval developmental delay caused by imidacloprid. We have included the above text in our revised manuscript. Please see the lines 432-442 in the revised manuscript.

1. If there is gut damage, is it restored in the adults? It is not clear from the methods which precise stage of larval development was used for gut preparations. That information is important because prior to pupation larvae defecate for the first time and undergo shedding of the gut lining. That could profoundly affect some of the results in case gut preparations for microscopy were made close to this stage. If no food residues are found in the gut of control larvae, does it mean that they are close to pupation? Could the apoptosis found in gut of exposed larvae be the natural shedding of gut lining prior to pupation? All these possibilities have to be discussed and authors should clarify the precise larval stage used in every assay.

Response: Thank you for your important comments. In this study, all samples used for the assay were larvae that had developed to 5-day-old after oral administration imidacloprid at 2-day-old. This is described in detail in the Materials and Methods. See lines 507, 517-521 in the revised manuscript. In general, 6-day-old bee larvae cease feeding and begin their first defecation at approximately 7-day-old. However, in our study, intestinal sections were prepared from 5-day-old larvae that had not fasted or defecated, when the intestinal mucosa was normal and not undergoing shedding. In this case, we found that imidacloprid caused damage to intestinal structures, apoptosis of intestinal cells, incomplete formation of the peritrophic membrane, and undigested food residues in the intestine. We believe that these results are objective and reliable.

1. Honeybee have 5 larval instars, not 7 (Figure 1). That creates confusion about which larval stage the authors used.

Response: Thank you very much for pointing out this editorial error, which we have corrected, please see Figure 1.

1. The Results section does not state the numbers by which parameters measures have changed, neither the values of significance. How much is the impact in growth index, body mass, gene fold change, etc?

Response: Thank you very much for pointing out this important problem. We have revised the Results section according to your suggestions. Please see the revised manuscript.

1. Mention figures in order (5c comes before 5b in the text)

Response: Thank you very much for the comment. We have revised according to your suggestions. Please see the lines 208-212 in the revised manuscript.

1. Paraquat is a herbicide not a pesticide

Response: Thank you for pointing out the loose wording. We have revised according to your suggestions. Please see the lines 316-319 in the revised manuscript.

1. What is the evidence that imidacloprid reduces growth index by inhibiting 20E? The authors provide real time data and discuss the data in terms of correlation. But correlation does not mean causation. Reduction in growth index could come from multitude of factors such as ROS affecting mitochondrial energy metabolism.

Response: We deeply appreciate your insightful comments and valuable suggestions. In this study, although we conducted an in-depth analysis of ecdysone regulation, which is crucial for insect larval development, and found some clues, as you pointed out, this is not the sole reason for larval developmental delay. In fact, animal growth and development are collectively regulated by numerous physiological, biochemical, and genetic factors. The the decline in the growth index may be due to other factors as you mentioned, such as oxidative stress impairing mitochondria, dysregulated neuro-endocrine axis caused by imidacloprid targeting neurons, poor nutrient absorption, impaired movement, etc, as animal growth and development are collectively regulated by numerous physiological, biochemical, and genetic factors. We have incorporated this understanding into the revised manuscript. Please see the lines 389-394 in the revised manuscript.

1. The authors state that "digestion and breakdown of nutrients is impaired by imidacloprid", the evidence discussed in the paragraph however supports only that imidacloprid impairs some of the genes involved in these processes.

Response: Thank you for your comments and valuable insights. In this paragraph, a lack of clarity and completeness in our writing may have led to the misconception that the evidence discussed only demonstrates the effects of imidacloprid on specific genes in these processes. In fact, our intent in this paragraph was to analyze and discuss the effects of imidacloprid on nutrient digestion and breakdown in larvae and to explore the causes of larval developmental delay. We demonstrated this using tissue sections, qRT-PCR and correlation analysis, which showed that the intestinal structure was disrupted and the expression of genes involved in nutrient digestion and catabolism was suppressed, resulting in defects in the catabolic utilization of food and consequently the presence of many food residues. In addition, there was a positive correlation between these genes and larval developmental delay. All this may be another important factor contributing to imidacloprid-induced larval developmental delay. We have revised and incorporate the above logic into the revised manuscript. Please see the lines 407-431 in the revised manuscript.

1. There is no evidence for the claim that overexpressing P450s and antioxidant enzymes cause a reduction in growth index. No transcriptome analysis was performed so it is unknown under the circumstances presented here how all the other P450s, antioxidant genes and overall gene profiles are responding. Surely, some genes will be repressed. Reduction in growth index could stem from, oxidative stress impairing mitochondria, dysregulated neuro-endocrine axis caused by imidacloprid targeting neurons, poor nutrient absorption, impaired movement, etc.

Response: Thank you for your comments and valuable insights. Indeed, as you have pointed out, drawing the conclusion that antioxidants and detoxification are significant contributors to larval developmental retardation solely based on correlation analysis is inherently flawed and lacks critical support, especially in the absence of P450 and antioxidant enzyme overexpression and comprehensive transcriptome analysis of other P450s, antioxidant genes, and the entire gene map. We have revised and included in the revised manuscript. Please see lines 461-467 in the red text in the revised manuscript. We have revised and incorporate the above logic into the revised manuscript. Please see the lines 407-431 in the revised manuscript.

1. How come the decreased ATP and glycogen levels have no effect on time to pupation? Extra time points for gene expression, measurements of gut damage, ATP levels, ROS, etc, are vital to answer how the exposed larvae eventually catch up with the unexposed group. Also, it is vital to understand whether these larval impacts translate to impacts on adults.

Response: We sincerely thank you for your insightful comments and suggestions! These important scientific issues you've raised are a good example of your high-level scientific thinking, and they will help us and other future researchers in the field to more comprehensively study and interpret the toxic effects of imidacloprid on honey bee larvae and their potential mechanisms, as well as the mechanisms of larval resistance and adaptation to imidacloprid. According to your comments, we will adapt our experiments and conduct more thorough research in the future to address the above issues.

1. I am confused about the author's definition of developmental rate; rate gives the notion of speed to achieve something. But the authors use developmental rate as a measure of viability (number of larvae that successfully pupated). There seems to be a significant decrease in their developmental rate plot (Fig 1i), but at the same time the authors show in Figure 1c (and mention throughout the manuscript) that there is no difference in probability of survival. This is quite confusing and the method section regarding these data is too concise and does little to help explain what the authors were trying to measure. The whole section on developmental traits would benefit of more details on how experiments were conducted and equipment used.

Response: Thank you so much for your valuable comments. Yes, as you can see, there appears to be a significant decrease in developmental rate but no difference in survival probability, which is an intriguing finding of this study. This finding suggests that the 377 ppb imidacloprid dose is not as harmful to the larvae as previously thought. Imidacloprid appeared to limit the larval ability to molt and develop only to a certain extent, but had no effect on the developmental process, let alone survival. It's worth investigating the underlying mechanism. As a result, we have included this question in the design of future studies. In addition, following your suggestion, we have revised the description of the material and methods in this section, including the experimental method in more detail. For more information, please see the revised manuscript, lines 530-541.

1. The authors should try to make it clear what percentage of exposed larvae become adults? I am confused because the plot called developmental rate might be trying to convey this message, but developmental rate and viability are very distinct traits. What is the difference, if any, in the time it takes for exposed larvae to become adults in comparison to non-exposed ones? Is there a difference in adult body weight? The answers to these last two questions are important to start understanding if the impacts of imidacloprid on larvae alimentation would still impact these same individuals once they become adults, i.e., would there be impacts for the colony and workers activity?

Response: Thank you very much for your insightful comments. Unfortunately, this is where the research falls short. Culturing larvae to adulthood in 24-well cell culture plates is a significant technical challenge that we have yet to overcome. As a result, the important questions you raise, such as what percentage of exposed larvae become adults? How does the time to adulthood differ (if at all) for exposed larvae versus non-exposed larvae? Is there a difference in adult weight? Do the effects of imidacloprid on larval feeding persist after these individuals reach adulthood? Does imidacloprid damage to larvae affect colony and adult activity? We do not have answers at this time. We are aware that answers to the above questions will help people better understand how serious the effects of imidacloprid environmental residues on honey bee larvae and adults, as well as bee colonies as a whole, are, and will draw sufficient attention to them. We intend to break through this technological bottleneck of culture larvae to adulthood in future studies and incorporate the above scientific questions into our next research design. Thank you again for your insightful comments! This gives us new research ideas.